# Isolation and Characterization of the *Arapaima gigas* Growth Hormone (ag-GH) cDNA and Three-Dimensional Modeling of This Hormone in Comparison with the Human Hormone (hGH)

**DOI:** 10.3390/biom13010158

**Published:** 2023-01-12

**Authors:** Eliana Rosa Lima, Renan Passos Freire, Miriam Fussae Suzuki, João Ezequiel Oliveira, Vanessa Luna Yosidaki, Cibele Nunes Peroni, Thaís Sevilhano, Moisés Zorzeto, Lucas Simon Torati, Carlos Roberto Jorge Soares, Igor Daniel de Miranda Lima, Thales Kronenberger, Vinicius Gonçalves Maltarollo, Paolo Bartolini

**Affiliations:** 1Instituto de Pesquisas Energéticas e Nucleares (IPEN-CNEN), Cidade Universitária, São Paulo 05508-000, SP, Brazil; 2Piscicultura Raça, Canabrava do Norte 78658-000, MT, Brazil; 3EMBRAPA Pesca e Aquicultura, Loteamento Água Fria, Palmas 77008-900, TO, Brazil; 4Departamento de Produtos Farmacêuticos, Faculdade de Farmácia, Universidade Federal de Minas Gerais (UFMG), Av. Presidente Antônio Carlos, 6627, Belo Horizonte 31270-901, MG, Brazil; 5Institute of Pharmacy, Pharmaceutical and Medicinal Chemistry and Tübingen Center for Academic Drug Discovery, Eberhard Karls University Tübingen, Auf der Morgenstelle 8, 72076 Tübingen, Germany; 6Department of Oncology and Pneumonology, Internal Medicine VIII, University Hospital Tübingen, Otfried-Müller-Straße 10, DE, 72076 Tübingen, Germany; 7Tübingen Center for Academic Drug Discovery & Development (TüCAD2), 72076 Tübingen, Germany; 8School of Pharmacy, Faculty of Health Sciences, University of Eastern Finland, 70211 Kuopio, Finland

**Keywords:** *Arapaima gigas*, growth hormone, pirarucu, molecular modeling, in silico sequencing

## Abstract

In a previous work, the common gonadotrophic hormone α-subunit (ag-GTHα), the ag-FSH β- and ag-LH β-subunit cDNAs, were isolated and characterized by our research group from *A. gigas* pituitaries, while a preliminary synthesis of ag-FSH was also carried out in human embryonic kidney 293 (HEK293) cells. In the present work, the cDNA sequence encoding the ag-growth hormone (ag-GH) has also been isolated from the same giant Arapaimidae Amazonian fish. The ag-GH consists of 208 amino acids with a putative 23 amino acid signal peptide and a 185 amino acid mature peptide. The highest identity, based on the amino acid sequences, was found with the Elopiformes (82.0%), followed by Anguilliformes (79.7%) and Acipenseriformes (74.5%). The identity with the corresponding human GH (hGH) amino acid sequence is remarkable (44.8%), and the two disulfide bonds present in both sequences were perfectly conserved. Three-dimensional (3D) models of ag-GH, in comparison with hGH, were generated using the threading modeling method followed by molecular dynamics. Our simulations suggest that the two proteins have similar structural properties without major conformational changes under the simulated conditions, even though they are separated from each other by a >100 Myr evolutionary period (1 Myr = 1 million years). The sequence found will be used for the biotechnological synthesis of ag-GH while the ag-GH cDNA obtained will be utilized for preliminary Gene Therapy studies.

## 1. Introduction

*Arapaima gigas* (the Brazilian pirarucu) is a giant fish of the order of Osteoglossiformes, which is native to the Amazon River basin that can reach 3 m in length and weigh up to 250 kg. It is very important for human nutrition and extractivism in the region, but this species is in danger of disappearing because of the increasing human presence and exploitation by predatory fishing [1,2,3]. Commercial breeding is still incipient and established reproductive centers face difficulties, especially due to the limited reproduction capacity of this species in captivity [4,5,6,7,8,9,10].

It is known that the two pituitary gonadotrophic hormones, the follicle-stimulating hormone (FSH) and the luteinizing hormone (LH), regulate reproductive processes such as gametogenesis and follicular growth in all vertebrates [11,12,13,14]. Given the extensive experience of our research group with human pituitary gonadotrophins, we have carried out the molecular cloning and characterization of *A. gigas* FSH and LH alpha- and beta-subunit cDNAs, together with a preliminary synthesis in HEK293 cells and a characterization of ag-FSH [15,16].

Growth hormones could be particularly important for *A. gigas* breeding since, in addition to the well-known effects on somatic growth and muscle mass increase, it also has immune and reproductive functions stimulating spermatogonial proliferation and accelerating the spermatogenic processes in fish [17,18]. In fact, it has been demonstrated that GH promotes testosterone and estradiol production in the catfish *Clarias batrachus,* exerting a reproductive role in this species [19]. For these reasons, we are currently collaborating with established *A. gigas* reproduction centers in order to carry out the molecular cloning and characterization of ag-GH cDNA.

A schematic image of *A. gigas* and the signaling pathway of hormones is presented in Figure 1.

For this purpose, we have employed a two-pronged approach: bioinformatics (“in silico”) to derive the ag-GH cDNA from the whole genomic sequence of pirarucu, available via the very comprehensive and useful work of Vialle et al. [20], together with the classical method based on pituitary extracted mRNA [16] to confirm the previously found sequence. Based on the bioinformatics approach, it is practically impossible to find specific 5′UTR or 3′UTR sequences. Since our vector constructions usually do not need the 5′UTR, we chose to work only with the 3′UTR, which is adequate for the detection of the poly(A) signal and poly(A) tail.

The cloning of the GH cDNA for different species of fish has been reported by some authors [21,22,23,24], while other authors also reported the related synthesis in different hosts [25,26,27,28,29,30].

According also to our phylogenetic studies [15,16], the fish GH sequences that are expected to have the highest percentages of identity with ag-GH are those related to *Anguilla anguilla and Acipenser baerii.*

Considering the importance that ag-GH has on somatic growth, muscular mass increase, and also on reproductive functions, our goal has been to obtain, for the first time, its cDNA sequence. This will be used for ag-GH biotechnological synthesis and preliminary gene therapy studies. As in our previous work on ag-FSH and ag-LH [16], we employed 3D modeling for ag-GH, permitting now a comparison of this structure with that of human GH, separated from ag-GH by an evolutionary period of more than 100 Myr [20,31,32].

**Figure 1 biomolecules-13-00158-f001:**
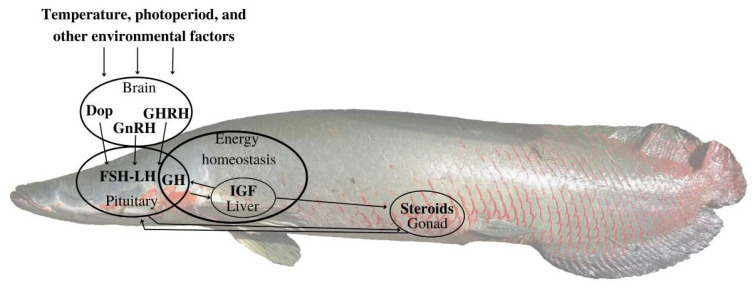
Schematic figure representing the main signaling pathways in the control of fish reproduction, adapted from Migaud et al. [33]. Environmental information (temperature, photoperiod, etc.), perceived in the brain (pineal, retina, and other receptors) initiates a reproductive cascade regulating the gonadotropin-releasing hormone (GnRH), which initiates the brain–pituitary–gonadal (BPG) cascade. While GnRH release stimulates FSH-LH release, dopamine (Dop) exerts an inhibitory effect. The growth hormone (GH) controlling the insulin-like growth factor (IGF) production in the liver is believed to be key a messenger in both energy homeostasis and gonadogenesis through the stimulation of gonadal sex steroids [19].

## 2. Materials and Methods

### 2.1. A. gigas Pituitary Gland Sources and the Detection of a First Partial Region of ag-GH

Pituitary glands were obtained from two fish farms. The first is located in São João da Boa Vista (São Paulo), and the processed material was used for studies regarding ag-LH and ag-FSH [16] and detection of a first partial region (~200 bp, “seq1”) of ag-GH, utilizing primer #1 and #2 (Table 1). Initially, the design of the two primers was based on conserved regions of the GH gene from three different fish species: *Anguilla anguilla*, *Cyprinus carpio*, and *Ictalurus punctatus*, available in GenBank under the accession numbers AY148493.1, M27000.1, and NM_001200245.1, respectively. For the selection of the species in this study, data previously obtained by our research group were used [16,17]. These GH sequences were aligned and the region that presented the closest profile for the three species was chosen as a safe region for the design of the primers used in this first step. The second pituitary collection was from Canabrava do Norte, Mato Grosso State, within the facilities of the Raça fish farm, collaborating with our project. The animals weighed 15–40 kg and the collected pituitary glands were 70–100 mg. These were stored in 1 mL of RNALater™ (ThermoFisher Scientific, Waltham, MA, USA), and transported to our laboratory in São Paulo in dry ice.

### 2.2. In Silico Prediction Analysis

After Sanger sequencing of “seq1”, it was used to identify the position of ag-GH in the *A. gigas* genome [20] (NCBI access number GCA_900497675.1) using MegaBLAST [34]. After having found a unique location (scaffold 1040) and employing the gene prediction tool Gnomon pipeline [35], four exons were rearranged according to Splign and Chainer, allocating one single variant of 627 nt, which included the whole coding sequence of ag-GH. ClustalΩ pairwise alignment [36] using 44 fish GH coding sequences provided a similarity ranging between 35.8 and 92.8 %, confirming our data. The fish species and respective GenBank accession numbers are listed below (Table 2).

### 2.3. Obtaining the Coding Region from the Pituitary Gland cDNA of A. gigas

After confirmation of the coding region in silico, new specific primers were designed using BLAST- NCBI, and a new PCR reaction was performed on pituitary extracted RNA, following the manufacturer’s guidelines. The Platinum™ SuperFi™ II DNA Polymerase system (Invitrogen™, ThermoFisher Scientific, Waltham, MA, USA) was used. A primer sense (#3) and another antisense (#4) (Table 1) were used, obtaining, according to the Sanger method, a fragment of approximately 600 bp that included the whole coding sequence of ag-GH. The latter perfectly matched the sequence previously obtained in silico.

### 2.4. Rapid Amplification of cDNA Ends (RACE) for Obtaining agGH 3′UTR

For rapid amplification of the 3′ ends of the cDNA, the commercial 3′ RACE System for rapid amplification of the cDNA ends kit (Invitrogen™, ThermoFisher Scientific, Waltham, MA, USA) was used. The primer for this reaction was also designed based on the sequence previously obtained in silico. From 5 μg of a “template strand” of total RNA extracted from the pituitary glands, a 3′ RACE reaction was performed using 1μL of PCR Anchor Primer (AP, #5, Table 1) (provided by the manufacturer) and 200U (1 µL) of the Go Script^®^ Reverse Transcription System (Promega, Madison, WI, USA) to obtain single-stranded cDNA. PCR was performed on single-stranded cDNA using 1 μL of 10 μM specific primer sense (#7) together with the AUAP antisense primer (#6) and 10 μL of 10 μM Taq polymerase Platinum SuperMix High Fidelity^®^ (Thermofisher Scientific, Waltham, MA, USA). Cycling was performed as per the manufacturer’s instructions. After completion of the reaction, the cDNA obtained was purified by agarose gel electrophoresis and sequenced.

### 2.5. Molecular Modelling

The initial three-dimensional (3D) models of *A. gigas* GH (ag-GH) were generated using the threading modeling method with a similar but updated protocol of previously reported work [16]. Five initial models were built using the SWISS-MODEL web server (https://swissmodel.expasy.org/, accessed on 26 June 2022) [37,38] based on the highest sequence identity with potential templates (PDB IDs: 3hhr [39], 1hwg [40], 1a22 [41], 1bp3 [42], and 1axi [43], with 49.72, 49.72, 49.16, 44.81, and 44.26% of identity, respectively). Subsequently, all five models were compared according to the global model quality estimate (GMQE) and QMEANDisCo global score [44], and the best-scored model was submitted to model refinement with DeepRefiner (http://watson.cse.eng.auburn.edu/DeepRefiner/, accessed on 6 August 2022) [45] and ModLoop (https://modbase.compbio.ucsf.edu/modloop/, accessed on 27 June 2022) [46] web servers. Additionally, a model obtained from AlphaFold [47,48] implemented on NMRbox [49] was also used to compare with other models. The refined models were also compared according to their validation metrics, such as the Q-MEAN scoring function [50] and Ramachandran plots [51] (Appendix A), and the best-performing model according to the quality control was submitted to molecular dynamics (MD) simulations.

### 2.6. Molecular Dynamics

The selected ag-GH model was submitted to an MD simulation and compared to the human GH (hGH) model simulations (PDB ID: 1hwg, the template employed) as the control. The protein structures were prepared using Epik [52] by adjusting the ionization states of the amino acid residues and further subjecting them to restrained energy minimization (PrepWiz, Maestro v2019.1). The missing residues of hGH were inserted manually following the available amino acid sequence of the PDB file. The N- and C-termini were capped, and ions and waters of crystallization were removed. Disulfide bonds were set automatically. The final structure was minimized using PrepWizard and the OPLS4 force field [53], with movements constrained to a maximum variation of 0.3 Å.

MD simulations were carried out using Desmond [54] with the OPLS4 force field [53]. The simulated system encompassed the refined protein model, a predefined explicit solvent model for water (TIP3P [55]) and counterions (Na^+^ or Cl^-^ adjusted to neutralize the overall system charge). The entire system was contained in a 13 Å^3^ cubic box with periodic boundary conditions. We used a time step of 1 fs and the short-range coulombic interactions were treated using a cut-off value of 9.0 Å, while the smooth particle mesh ewald method (PME) handled the long-range coulombic interactions [56]. Initially, the system was minimized using steepest descent followed by equilibration. All simulations were run in an NPT ensemble (T = 310 K, Nosé–Hoover method; *p* = 1.01325 bar, Martyna–Tobias–Klein method) with default Desmond settings. Reversible reference system propagator algorithms (RESPA) with 2 fs, 2 fs, and 6 fs timesteps were used for bonded, near and far, respectively. A constant temperature of 310 K was maintained throughout the simulation using the Nosé–Hoover thermostat algorithm, together with the Martyna–Tobias–Klein Barostat algorithm to maintain 1 atm of pressure. After minimization and relaxation of the system, we proceeded with five independent production steps of 200 ns (total 1 μs) for both systems, with randomly generated seeds. The concatenated trajectory of the five replicates for each system was analyzed according to RMSD and RMSF calculations for C-alpha atoms and secondary structure formation. In this sense, RMSD plots were generated using a single concatenated trajectory of 1 μs represented by five replicas of 200 ns, while RMSF was represented by the average values calculated for each amino acid residue. Finally, clusterization of structures was performed to retrieve the most frequent conformation of both proteins.

### 2.7. Quantitative Similarity Comparison between the ag-GH Model and the Template (hGH)

Two distinct methods were employed to structurally compare the generated model and the selected template: ConSurf (https://consurf.tau.ac.il/consurf_index.php, accessed on 1 September 2022) [57,58,59] and 3D visualization of sequence similarity (SS3D) [60]. The ConSurf method calculates the evolutionary rate of each amino acid position of the protein to identify the most structurally and functionally important positions. In this context, the evolutionary rate is inversely proportional to the degree of conservation of a given site. The most populated cluster frames from the MD of the ag-GH and the hGH models were used as input.

The SS3D method was carried out by comparing the ag-GH and hGH clusters from the MD results to the template crystallographic structure (PDB ID: 1hwg) to highlight any difference between these homologous proteins by combining sequence and structure information. At this step, a contact map of the α-carbons was generated with a cut-off distance of 0.6 nm. The SS3D value for each of the selected contacts was calculated employing the experimental structure of hGH (PDB ID: 1hwg) as the reference for comparing the structures. For this purpose, we selected residues whose α-carbon was within a radial distance of 1 nm from any of the residues in contact, considering only residues that persisted within this radius for more than 75% of the frames. Finally, the structures were mapped between 0 (dissimilar) and 1 (similar), according to the calculated SS3D value.

## 3. Results

### 3.1. cDNA Sequencing

The DNA sequences obtained in silico and via 3′ RACE using RNA from two different pituitary glands were compared and aligned. The *A. gigas* GH cDNA sequence (Figure 2) was 1061 bp in total length, with an open reading frame of 627 bp, beginning with the first ATG codon at position 1 bp and ending with the stop codon at position 627, presenting a 434 bp 3′-UTR. A polyadenylation signal (GATAAA) was recognized 40 bp upstream from the poly(A+) tail. The coding region translates into a polypeptide of 208 amino acids, while the cleavage site for the putative signal peptide was between amino acids 23 and 24. The signal peptide prediction was based on Signal P 6.0 [61].

This corresponds to a mature peptide of 185 and a signal peptide of 23 amino acids. The mature peptide of ag-GH thus contains four conserved cysteines and six prolines (Pro^2^, Pro^34^, Pro^55^, Pro^5^^7^, Pro^85^, and Pro^135^), three of which (underlined) are highly conserved (Figure 2). There is one potential glycosylation site at Asn^182^ that can be seen in all species, but one, in the alignment of the ag-GH mature peptide with the forty-one teleosts and three Acipenseriformes in Figure 3.

The comparison between the amino acid sequences of GH in 14 orders of fish provided the values reported in Table 3. The highest identity for ag-GH was with Elopiformes (82.0%), followed by Anguilliformes (79.7%) and Acipenseriformes (74.5%), while the lowest was with Pleuronectiformes (53.8%) and Gadiformes (62.3%). It should be noted, though, that these percentages are influenced by the number of species studied and by those that could be compared with ag-GH. It is interesting to observe that the percent identity between ag-GH and human-GH was 44.7%, while that between hGH and mouse-GH was 66.1% (Table 4). This will be discussed later in the context of the potential for in vivo bioactivity testing.

### 3.2. Three-Dimensional Model of agGH

The five initial models obtained from the SWISS-MODEL were very similar. The model generated using the structure of hGH determined by X-ray crystallography (PDB ID: 1hwg) as a template was chosen due to the slightly higher GMQE score. Two refined models were then obtained from the DeepRefiner and ModLoop servers. The model retrieved from the first method drastically improved the quality of the model concerning the Ramachandran plot (0% of outliers) and QMEAN Z-scores (Appendix A). In addition, the refined model was compared with the template to validate several of the structural features before the MD simulations. Finally, the model obtained with AlphaFold improved the quality of Cβ atoms, all atoms, and solvation scores, but decreased the QMEAN Z-score and torsion score. Therefore, we followed the refined model from the SWISS-MODEL. The 3D structures of the selected model and the experimental template structure are very similar (Figure 4A). All secondary structures were conserved, and their superimposition shows low dissimilarity (RMSD 0.81 Å, where <1 is considered similar). Furthermore, the electrostatic surfaces of the two proteins were also similar with comparable charge distributions, in particular in the regions reported to bind the GH receptor (regions I, II, and III in Figure 4B). Therefore, the quality measurements and the structural comparison both suggested that the model selected was suitable for further analysis.

The subsequent MD simulations of ag-GH and hGH displayed similar structural stability along the simulation timescale (Figure 4C), as well as the fluctuations of the alpha carbons of both proteins (Figure 4D). Taken together, these results suggest that the two proteins have similar structural properties, without major conformational changes under the simulated conditions. The most representative conformations of ag-GH and hGH maintained overall folding and secondary structures that were similar to those of the X-ray structure of hGH employed as the template (Figure 4E,F). Therefore, the proposed model for ag-GH appears to be stable, indicative of the quality of the model generated by this procedure. Indeed, the MD simulations only promoted structure relaxation rather than causing major conformational changes between the refined model of ag-GH and the initial X-ray structure of hGH. Moreover, the two disulfide bonds of ag-GH and hGH present the same 3D position in the two proteins (Figure 4G).

Lastly, we also analyzed the behavior of residues reported to bind to the GH receptor [40] in both models: Lys41, Gln46, Ser51, Glu56, Ser62, Lys64, Arg167, and Lys168 from *h*GH and Arg38, Asp42, Ala47, Asp52, Thr58, Arg60, Lys160, and Lys161 from *ag*GH (Figure 5). Interestingly, the residues were conserved in terms of sequence alignment (Figure 5A) and according to the three-dimensional alignment as well (Figure 5B). In addition to the electrostatic potential surfaces that suggested a similar ability to bind to their receptor, the solvent-accessible surface area distribution of the above-mentioned residues was also very similar along the entire MD simulation (Figure 5C), suggesting the availability to interact with their receptor binding site instead of being buried.

### 3.3. Evolutionary Implications and Comparison of the Three-Dimensional Contacts of ag-GH and hGH Models

An independent comparison of the amino acid sequence conservation and the 3D similarity (Figure 6) confirmed the previous results. Thus, human (Figure 6A) and *A. gigas* (Figure 6B) GH models displayed similar amino acid conservation profiles in comparison with homologous proteins. Furthermore, the regions with the highest conservation scores were located at α-helices that interact with the GH receptor (Figure 4B). C-terminal residues could not be classified due to insufficient data.

The same similarity profile was found in the SS3D studies. Both models were very similar to the experimental hGH structure of the template, in particular considering their α-helices (Figure 6F,G). In this regard, the contact matrices were very similar (Figure 6C,D) and the hGH model exhibited greater similarity to the hGH experimental structure (Figure 6E), as would be expected.

## 4. Discussion

For the first time, ag-GH cDNA has been sequenced and characterized, and also an *A. gigas* hormone was sequenced via in silico methods, and this sequence was compared with that obtained via the classical method. As far as we know, two homologous bioactive molecular structures, at an evolutionary distance of >100 million years, were never compared by three-dimensional modeling and molecular dynamics.

The characterization of ag-GH cDNA points to a protein with a putative signal peptide of 23 amino acid residues and a mature hormone of 185 amino acids. There are four cysteines (Cys^49^, Cys^158^, Cys^175,^ and Cys^183^) that are perfectly conserved and located at nearly the same position as those present in the GH of other vertebrates. One potential N-glycosylation site (Asn–Cys–Thr) is present at Asn182, separated from the stop codon only by a Leu. Although glycosylation is not usually found in GH molecules, it would be interesting to see whether ag-GH can be glycosylated in other eukaryotes. The same C-terminal (Asn–Cys–Thr–Leu) is very common in the GH cDNA of many species of fish, such as salmon [21], Japanese eel [26], pacu, giant catfish, channel catfish, bighead carp, silver carp [24,29] and prenant’s schizothoracin [30]. As a matter of fact, we can deduce from Figure 2 that only one species (*Pimephales promelas*) out of the forty-five considered does not have the same N-terminal ending.

The two disulfide bonds that can be formed by the four Cys are critical for the integrity and biological activity of the hormone. Consequently, the C-terminal region, where three out of four cysteines are located, should be particularly important for GH structure and function [62].

Six prolines, three of which are highly conserved, confirm that they can be essential for the protein structure since they tend to bend the local amino acid alignment and, therefore, fold the protein [16,63].

Table 3, comparing the pairwise identity of GH peptides between fish orders, shows that GH (38–89%) is less variable than FSHβ (28–67%) and approximately as variable as LHβ (47–89%) [16]. The greater ag-GH identity with Elopomorpha confirms our previous data obtained with ag-GTHα, ag-LHβ, and ag-FSHβ [15,16].

It is noteworthy that ag-GH exhibits a 44.7% sequence identity with the corresponding human hormone (hGH), and a 57.8% identity with a mouse GH. These values suggest the possibility of bioactivity determination via the classical in vivo bioassay based on dwarf “little” mice (lit/lit) [64], a hypothesis that still needs to be verified experimentally using purified ag-GH against the WHO’s hGH International Standard. If this can be confirmed, it would be useful to set up an ag-GH standard for inter-laboratory comparisons and future production and purification. Even though immunological activity does not reflect in vivo biological activity, we found that a Western blot, based on anti-hGH antiserum, worked perfectly using ag-GH as an antigen (data not presented).

The theoretical protocol consisted of homology modeling followed by molecular dynamics simulations using multiple replicas with long timescale (200 ns or more) simulations to ensure that the final results were representative in terms of folding and structural motions [65,66,67]. Furthermore, in the simulations, both systems (ag-GH and hGH as control) presented similar conformational movements, as indicated by the RMSD and RMSF values (Figure 4C,D). The observed minor differences in the RMSD were not unexpected since different models of the same protein can display RMSD differences during long simulations [68].

The analysis of protein structures in terms of the most populated cluster (Figure 4E,F) indicated that ag-GH and hGH have very similar foldings, even after structural relaxation during the molecular dynamics simulation. In addition, the similarity of the more important structural features of the protein, such as the receptor binding regions and the disulfide bonds (Figure 4B,G, respectively), suggests that the functionality of the two proteins is conserved [69,70,71].

The function, interaction, and folding of proteins depend on a select group of specific amino acids and these tend to be conserved throughout evolution or replaced by residues with similar biophysical characteristics that still preserve their biological role [72]. Therefore, in order to confirm the consistency of the structures generated, the trajectories obtained from the MDs were analyzed by two different approaches.

The first analysis performed was to assess the evolutionary similarity through phylogenetic relationships between homologous sequences. Indeed, both hGH and ag-GH were found to have a very similar conservation profile when compared to homologous proteins, with conservation regions that coincided with each other (Figure 6A,B). Although there was only approximately a 50% sequence similarity between hGH and ag-GH, the most conserved region coincides exactly with the site responsible for the interaction with the GH receptor. As previously shown by other works, this result is evidence that proteins share their functional roles [58,73].

To visualize details regarding the three-dimensional similarity and differences of the structures, the second approach analyzed a combination of sequence and structural similarity. Thus, ag-GH and hGH showed an SS3D value similar to that of the crystallographic structure of hGH (Figure 6E), which indicates that the two protein models have comparable residue identities around their α-carbon contacts. The SS3D tool indicates important sites involved in carrying out the function of homologous proteins, with these regions showing a high degree of spatial similarity [60]. Again, the region of greatest similarity corresponded to the site of interaction with the GH receptor, further strengthening the evidence that the proteins share a common function (Figure 6F,G).

Taken together, the results (predicted fold, electrostatic potential surface, evolutionary similarity, and 3D similarity) reinforce our hypothesis that the similarity of the proteins is related not only to their 3D structures, but also to their functionality as well.

In addition to the novelty of having cloned and sequenced ag-GH for the first time, there are practical aspects related to our ongoing collaboration with the two pirarucu fishing stations. Since fish GH is known for its effects on somatic growth and muscle mass increase and its action on reproduction, we are developing a preliminary gene therapy approach. A practical limitation is that repeated injections highly stress and decrease the appetite of these fish, so usual modes of GH administration are not really recommended. We will, therefore, study the effects of i.m. administrations of ag-GH cDNA, injecting only once a month or even less often, on ag-GH endogenous levels and mass increase, as our research group studied in dwarf “little” mice [74,75,76], and little/scid mice [77,78].

## Figures and Tables

**Figure 2 biomolecules-13-00158-f002:**
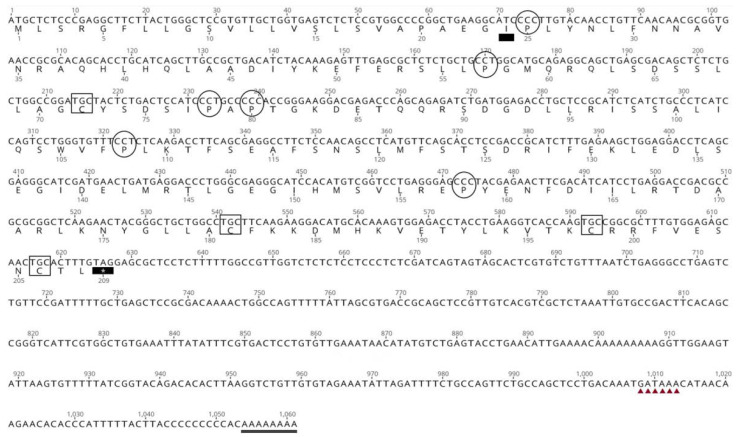
*Arapaima gigas* growth hormone cDNA and amino acid sequence. Signal peptide prediction indicates the presence of a cleavage site between residues 23 and 24. A black rectangle indicates the first amino acid of the mature peptide (I) and the asterisk is the stop codon; C, inside a square, indicates the cysteines involved in disulfide bridges; and P, inside a circle, indicates proline residues. Triangles indicate the poly-A signal (GATAAA) followed by the poly-A tail. GenBank accession number OP575308.

**Figure 3 biomolecules-13-00158-f003:**
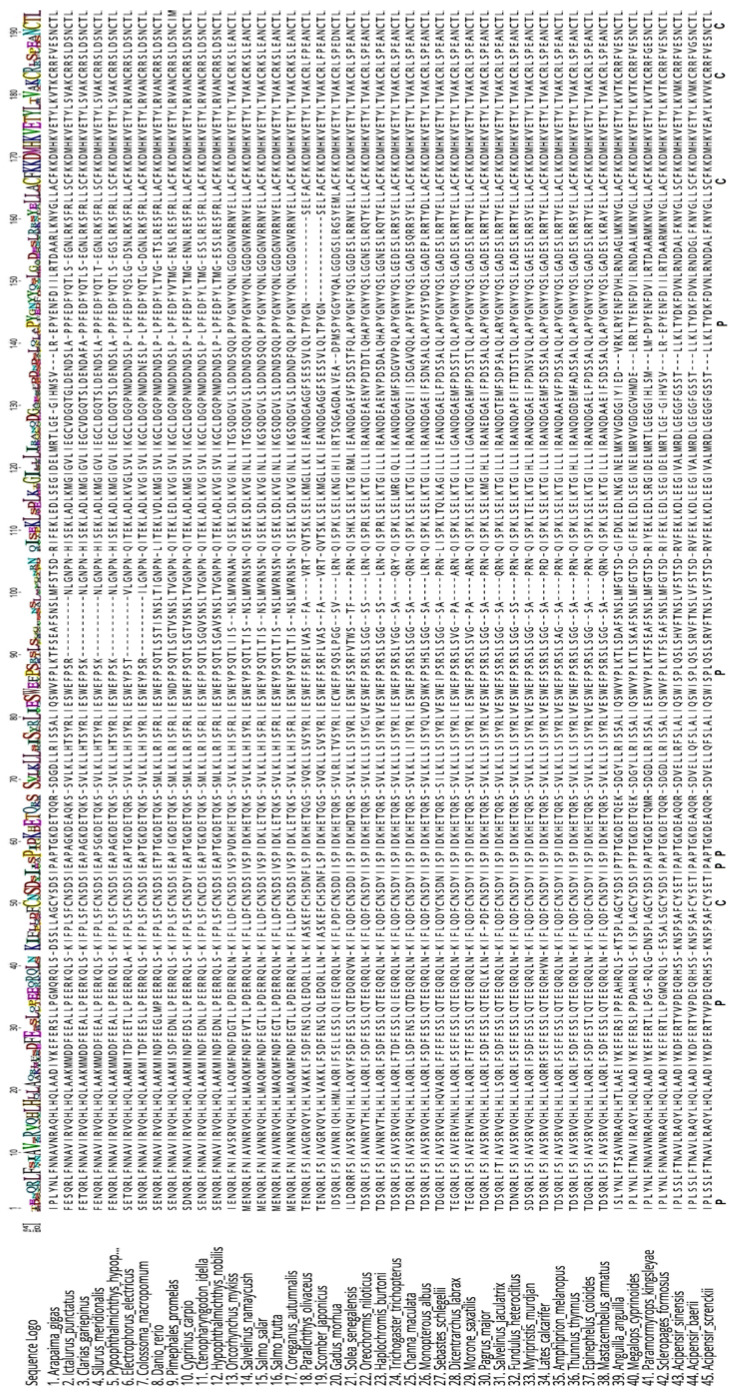
Alignment of the GH cDNA coding sequence from 45 fish species, including *Arapaima gigas*. The symbols indicate prolines (P) and cysteines (C).

**Figure 4 biomolecules-13-00158-f004:**
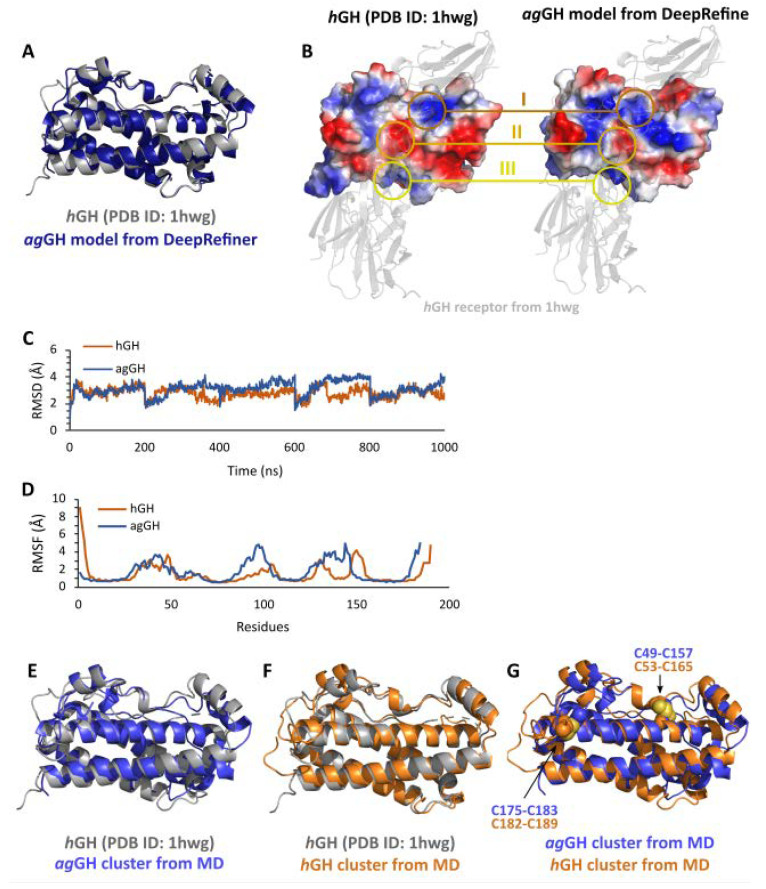
Superimposition of the template (PDB ID: 1hwg) and the refined structure (**A**). A comparison of the electrostatic potential surfaces (ranging from red, the electron-rich region, to blue, the electron-deficient region) of the template structure with that of the refined initial ag-GH model and with the PDB structure of the hGH receptor, shown as a transparent grey cartoon (**B**). RMSD (**C**) and RMSF (**D**) values were calculated for C-alpha atoms during the whole trajectories for both the ag-GH model (dark blue) and the hGH model (dark orange). The RMSD plot is presented as five subsequentially concatenated 200 ns replicas resulting in a single plot of 1 μs. Superimposition of the ag-GH model (blue) retrieved from the cluster analysis of the MD simulations with the experimental template structure (grey) (**E**). Superimposition of the hGH model (orange) retrieved from the cluster analysis of the MD simulations with the experimental template structure (grey) (**F**). Superimposition of the ag-GH model (blue) and the hGH model (orange) retrieved from the cluster analyses of the MD simulations. The two disulfide bonds are displayed in sphere representation (**G**).

**Figure 5 biomolecules-13-00158-f005:**
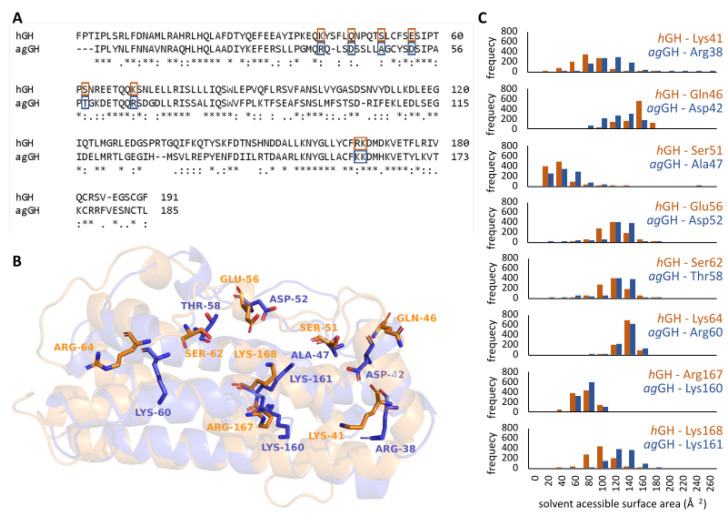
Alignment of hGH and agGH sequences using Clustal Omega [36], an * (asterisk) indicates positions which have a single, fully conserved amino acid residue, (colon): indicates residues with strongly similar porperties, and (period) indicates residues with weakly similar properties (**A**). Alignment of three-dimensional structures highlighting the residues reported to be involved in binding at the receptor’s binding site (**B**). Histograms of calculated solvent-accessible surface area distribution of each amino acid residue from both models during MD simulations (**C**).

**Figure 6 biomolecules-13-00158-f006:**
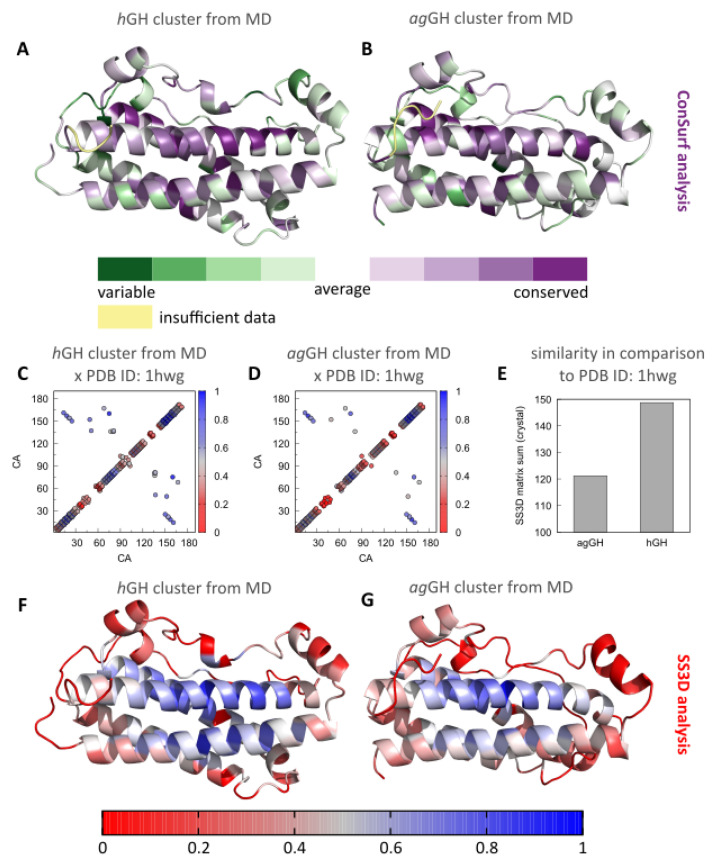
ConfSurf maps for the hGH model (**A**) and the ag-GH model (**B**) display the highly conserved regions of the proteins (purple) and the highly variable amino acid sites (green). Contact maps from the SS3D analysis for the hGH model (**C**) and the ag-GH model (**D**), as well as the overall SS3D similarity in comparison with the experimental template structure (PDB ID: 1hwg) (**E**), and the maps from the SS3D similarity analysis (**F**,**G**).

**Table 1 biomolecules-13-00158-t001:** Primers used in the cloning of *A. gigas* growth hormone.

Number	Direction	Name	Sequence
Primer 1	Sense	ag-GH1	5′ TCA GCA AGA CCT CCC CA 3′
Primer 2	Antisense	ag-GH2	5′ TTG AAG CTG CAT GTG GGA TCC TTG 3′
Primer 3	Sense	GSP1	5′ GAG GCT TCT TAC TGG GCT CC 3′
Primer 4	Antisense	GSP6	5′ AAG TGC AGT TGC TCT CCA CAA 3′
Primer 5	Antisense	AP	5′ GGC CAC GCG TCG ACT AGT ACT TTT TTT TTT TTT TTT T 3′
Primer 6	Sense or Antisense	AUAP	5′ GGC CAC GCG TCG ACT AGT AC 3′
Primer 7	Sense	GSP8	5′ CTC TCA AGA CCT TCA GCG AGG 3′

**Table 2 biomolecules-13-00158-t002:** Fish species and GenBank accession numbers of the sequences used in the amino acid percentage determinations and cDNA cloning of the fish growth hormone.

Organism	Nucleotide (mRNA)	Amino Acid	Superorder, Order, and Family
*Arapaima gigas*			Osteoglossomorpha, Osteoglossiformes, Arapaimidae
*Acipenser baerii*	JX947839.1	AFX71645.1	Chondrostei, Acipenseriformes, Acipenseridae
*Anguilla Japonica*	M24066.1	AAA48535.1	Elopomorpha, Anguilliformes, Anguillidae
*Anguilla anguilla*	AY148493.1	AAN61122.1	Elopomorpha, Anguilliformes, Anguillidae
*Chanos chanos*	XM_030794045.1	XP_030649905.1	Ostariophysi, Cypriniformes, Cyprinidae
*Colossoma macropomum*	XM_036564141.1	XP_036420034.1	Ostariophysi, Characiformes, Serrasalmidae
*Coregonus clupeaformis*	XM_041852232.2	XP_041708166.2	Protacanthopterygii, Salmoniformes, Salmonidae
*Electrophorus electricus*	XM_027026729.2	XP_026882530.1	Ostariophysi, Gymnotiformes, Gymnotidae
*Lates calcarifer*	X59378.1	CAA42022.1	Percoidei, Perciformes, Latidae
*Megalops cyprinoides*	XM_036552661.1	XP_036408554.1	Elopomorpha, Elopiformes, Megalopidae
*Myripristis murdjan*	XM_030057829.1	XP_029913689.1	Berycoidei, Beryciformes, Holocentridae
*Pangasianodon hypophthalmus*	JF303887.1	ADX99243.1	Ostariophysi, Siluriformes, Pangasiidae
*Paramormyrops kingsleyae*	XM_023835741.1	XP_023691509.1	Osteoglossomorpha, Osteoglossiformes, Mormyridae
*Scleropages formosus*	XM_018764341.2	XP_018619857.1	Osteoglossomorpha, Osteoglossiformes, Osteoglossidae
*Salmo trutta*	XM_029727357.1	XP_029583217.1	Protacanthopterygii, Salmoniformes, Salmonidae
*Salmo salar*	X14305.1	CAA32481.1	Protacanthopterygii, Salmoniformes, Salmonidae
*Salvelinus namaycush*	XM_038989124.1	XP_038845052.1	Protacanthopterygii, Salmoniformes, Salmonidae
*Toxotes jaculatrix*	XM_041062256.1	XP_040918190.1	Acanthopterygii, Perciformes, Toxotidae
*Salvellinus alpinus*	XM_024010963.1	XP_023866731.1	Protacanthopterygii, Salmoniformes, Salmonidae
*Monopterus albus*	AY265351.1	AAP42271.1	Acanthopterygii, Synbranchiformes, Synbranchidae
*Sebastes schlegelii*	U89917.1	AAB49492.1	Acanthopterygii, Scorpaeniformes, Sebastidae
*Solea senegalensis*	U01143.1	AAA60372.1	Acanthopterygii, Pleuronectiformes, Soleidae
*Paralichthys olivaceus*	D29737.1	BAA06159.1	Acanthopterygii, Pleuronectiformes, Paralichthyidae
*Thunnus thynnus*	X06735.1	CAA29914.1	Acanthopterygii, Perciformes, Scombridae
*Scomber japonicus*	GU138643.1	ACZ04987.1	Acanthopterygii, Perciformes, Scombridae
*Chrysophrys major*	AB904715.1	BAR88081.1	Acanthopterygii, Perciformes, Sparidae
*Epinephelus coioides*	AY513647.1	AAR97729.1	Acanthopterygii, Perciformes, Serranidae
*Morone saxatilis*	XM_035674082.1	XP_035529975.1	Acanthopterygii, Perciformes, Moronidae
*Dicentrarchus labrax*	GQ918491.1	ADB23477.1	Acanthopterygii, Perciformes, Moronidae
*Amphiprion melanopus*	HM135406.1	ADJ57589.1	Acanthopterygii, Perciformes, Pomacentridae
*Oreochromis niloticus*	KT387598.1	AMB21545.1	Acanthopterygii, Perciformes, Cichlidae
*Acipenser screnckii*	KC460212.2	AGI96360.2	Chondrostei, Acipenseriformes, Acipenseridae
*Acipenser sinensis*	EU599640.2	ACC86123.1	Chondrostei, Acipenseriformes, Acipenseridae
*Ctenopharyngodon idella*	M27094.1	AAA58724.1	Ostariophysi, Cypriniformes, Cyprinidae
*Danio rerio*	AJ937858.1	CAI79040.1	Ostariophysi, Cypriniformes, Cyprinidae
*Hypophthalmichthys nobilis*	GQ497221.1	ADD62434.1	Ostariophysi, Cypriniformes, Cyprinidae
*Pimephales promelas*	AY643399.1	AAT91088.1	Ostariophysi, Cypriniformes, Cyprinidae
*Clarias gariepinus*	FJ823972.1	ACN97175.1	Ostariophysi, Siluriformes, Clariidae
*Ictalurus punctatus*	EU009499.1	ABS70461.1	Ostariophysi, Siluriformes, Ictaluridae
*Silurus meridionalis*	AF530481.2	AAP82934.1	Ostariophysi, Siluriformes, Siluridae
*Coregonus autumnalis*	X77245.1	CAA54461.1	Protacanthopterygii, Salmoniformes, Salmonidae
*Oncorhynchus mykiss*	NM_001124689.1	NP_001118161.1	Protacanthopterygii, Salmoniformes, Salmonidae
*Gadus morhua*	EU676171.1	ACD46080.1	Paracanthopterygii, Gadiformes, Gadidae
*Fundulus heteroclitus*	XM_012872083.3	XP_012727537.1	Acanthopterygii, Cyprinodontiformes, Fundulidae
*Trichogaster trichopterus*	AF157633.1	AAG60346.1	Acanthopterygii, Perciformes, Osphronemidae

**Table 3 biomolecules-13-00158-t003:** Percentage identity of GH peptides between fish orders.

	Superorder	Order	1	2	3	4	5	6	7	8	9	10	11	12	13	14
1.	Osteoglossomorpha	Osteoglossiformes (3)	86.4	53.8	62.7	63.1	63.2	74.5	82.0	79.7	62.6	68.2	66.3	65.6	62.3	70.8
2.	Acanthopterygii	Pleuronectiformes (2)	-	63.7	68.1	67.6	68.3	55.7	45.7	45.2	63.0	68.5	53.5	80.7	59.5	78.0
3.	Acanthopterygii	Scorpaeniformes (1)		-	100	84.8	84.3	66.0	40.1	40.1	72.6	78.2	52.4	84.9	65.7	86.3
4.	Acanthopterygii	Synbranchiformes (1)			-	100	86.3	67.2	40.1	40.1	72.1	77.7	52.9	84.9	66.7	86.5
5.	Berycoidei	Beryciformes (1)				-	100	66.9	41.5	41.0	74.1	79.5	55.8	84.7	71.6	87.4
6.	Chondrostei	Acipenseriformes (3)					-	97.2	81.5	79.4	62.2	66.6	69.0	64.8	65.7	68.4
7.	Elopomorpha	Elopiformes (1)						-	100	86.6	69.5	78.6	49.1	75.5	38.4	82.8
8.	Elopomorpha	Anguilliformes (1)							-	100	68.0	78.1	46.7	75.5	37.9	82.7
9.	Ostaryophysi	Siluriformes (3)								-	95.2	82.7	88.8	71.0	71.9	78.8
10.	Ostaryophysi	Cypriniformes (5)									-	92.9	89.2	68.5	77.6	78.3
11.	Ostaryophysi	Characiformes (1)										-	100	78.3	49.0	87.2
12.	Acanthopterygii	Perciformes (9)											-	84.8	62.3	70.8
13.	Paracanthopterygii	Gadiformes (1)												-	100	85.6
14.	Protacanthopterygii	Salmoniformes (6)													-	97.0

**Table 4 biomolecules-13-00158-t004:** Percentage identity of GH peptides of different species relative to *A. gigas.*

Common name	*Scientific Name*	Identity (%)	GenBank/NCBI
Pirarucu	*Arapaima gigas*	100.0	OP575308
Crocodile	*Crocodylus porosus*	61.5	AAB34999.1
Manatee	*Trichechus manatus*	60.9	XP_004374274.1
Australian parakeet	*Melopsittacus undulatus*	59.4	XP_005140349.2
Whale	*Balaenoptera physalus*	58.9	CAH42020.1
Dolphin	*Tursiops truncatus*	58.9	XP_033703592.1
Orca	*Orcinus orca*	58.9	XP_004275687.2
Mouse	*Mus musculus*	57.8	AAH61157.1
Ox	*Bos taurus*	56.3	AAX07134.1
Chimpanzee	*Pan troglodytes*	44.7	AAL72284.1
Human	*Homo sapiens*	44.7	KAI2584577.1

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
