# Peer review of "Isolation and Characterization of the Arapaima gigas Growth Hormone (ag-GH) cDNA and Three-Dimensional Modeling of This Hormone in Comparison with the Human Hormone (hGH)"

_biomolecules, 2023, doi:10.3390/biom13010158_

Round 1
Author Response
The Authors thank the Reviewers for the interesting and very useful suggestions that contributed to a substantial improvement of the manuscript.
The manuscript submitted by Lima et al., characterizes the sequence and structure of Arapaima gigas growth hormone (ag-GH) in comparison to human hormone (hGH). The authors followed best computational approaches to model the ag-GH hormone. I accept this publication with minor revisions. Please address the following comments.
1) Line 193, please change Nose-Hoover —--> Nosé–Hoover
ANSWER: Done.
2) In the Figure 3C panel, the RSMD plot is misleading as it is giving the impression of a single long run of 1 microsecond instead of 5 independent 200 ns runs. It would be great if authors could plot the avg and std deviation from each of the runs and plot the comparison between the ag-GH and GH proteins.
ANSWER: Thank you for the comment. The methodology of using independent replicas as a strategy to extend the computational timescale and sampling followed state-of-the-art references in the literature. In this sense, each frame of each replica is not equivalent to the counterparts of the other replicas. Therefore, calculating th average and standard deviation at each frame to plot them as a "single" 200 ns run could lead to misinterpretation of the results. Indeed, the RMSD plot mentioned did give the wrong impression and we have therefore changed both the main text and the figure caption to avoid any misunderstanding.
3) Is RMSD and RMSF calculated for the entire protein or C-alpha atoms. This is not clear in the text.
ANSWER: Thank you for the suggestion. We changed the figure caption and main text to make this point clear to the readership.
4) Did authors try to use Alpha-fold or other algorithms that are widely used in the community to predict the 3D structure of ag-GH protein and compare it with the model obtained from SWISS-modeller?
ANSWER: After your request, we built the agGH model with AlphaFold and compared it with the model previously obtained. No major improvements were observed in the overall quality of the model.
5) Apart from the overall structure, it would be great if authors could describe the behavior of functional site residues and their interactions in both the protein from the MD trajectories.
ANSWER: Thank you for the suggestions. Indeed, we calculated properties of residues reported to be involved in binding in the literature.

Reviewer 2 Report
Lima et al identified a cDNA supposed to encode for Arapaima gigas growth hormone and predict the structure of ag-GH using molecular modelling approach. Though the study might be valuable as suggested in the introduction the approaches (amplification of cDNA and rest of the work is based on in silico) employed are not novel and strong enough to allow me to accept this paper for publication in Biomolecules. However I have several comments that I hope will help the authors to improve the quality of the paper and I suggest this paper may be resubmitted to Fish Journal MDPI
1. The title indicate you have cloned ag-GH but no detail on the cloning is provided rather what the authors did is the identification of cDNA encoding for ag-GH by PCR and sequencing.
2. Line 98-100 the design of the two primers was based on conserved regions of the GH gene from three different fish species: Anguilla anguilla, prinus carpio, and Ictalurus punctatus, available in the GenBank under the accession 100 numbers AY148493.1, M27000.1, and NM_001200245.1, Why did you particularly chosen this species and the evolutionary relationship between these species and Arapaima gigas?
3. In which chromosome ag-GH gene is located based on your prediction and could please provide a chromosomal and gene map depicting the promoter region, intron and exons?
4. The figure1 showing the sequence of cDNA is not of high resolution, its resolution should be improved.
5. The same problem of very low resolution is applied to figure 2
6. Table 4 is not clear what is included on it and why identity values repeated twice and of what the identity is? Perhaps the title of the table created a confusion
7. Did you express this cDNA in ecoli and check its molecular weight and western blot of ag-GH? Did the same approach also applied to extract this hormone from pituitary glands?
8. Can you provide the agarose gel electrophoresis of both UTR and full length cDNA in the paper?
9. After amplification of whole coding sequences did you confirm the sequences with Sanger or NGS tool methods?
10. Line 91. Whats 100 MyR?
11. The paper need to be checked carefully for typos and grammatical error.
Author Response
The Authors thank the Reviewers for the interesting and very useful suggestions that contributed to a substantial improvement of the manuscript.
Reviewer #2
Lima et al identified a cDNA supposed to encode for Arapaima gigas growth hormone and predict the structure of ag-GH using molecular modelling approach. Though the study might be valuable as suggested in the introduction the approaches (amplification of cDNA and rest of the work is based on in silico) employed are not novel and strong enough to allow me to accept this paper for publication in Biomolecules. However I have several comments that I hope will help the authors to improve the quality of the paper and I suggest this paper may be resubmitted to Fish Journal MDPI
- The title indicate you have cloned ag-GH but no detail on the cloning is provided rather what the authors did is the identification of cDNA encoding for ag-GH by PCR and sequencing.
Answer: The title indicates that we have cloned ag-GH-cDNA as we wrote in previous work (Sevilhano et al. ref. 16). But “isolation” may be better as we wrote in other work (Faria et al. 2013, ref.15)
- 2. Line 98-100 the design of the two primers was based on conserved regions of the GH gene from three different fish species: Anguilla anguilla, Cyprinus carpio, and Ictalurus punctatus, available in the GenBank under the accession 100 numbers AY148493.1, M27000.1, and NM_001200245.1, Why did you particularly chosen this species and the evolutionary relationship between these species and Arapaima gigas?
Answer: We based the design of the first two primers on conserved regions of the GH gene of Anguilla anguilla, , Cyprinus carpio and Ictalurus punctuatus because the order of Anguilliformes, Cipriniformes and Siluriformes presented the highest percentages of identity with Osteoglossiformes, for the gonadotrophic hormones FSH and LH (Sevilhano et al., ref.16, Table 3)
- 3. In which chromosome ag-GH gene is located based on your prediction and could please provide a chromosomal and gene map depicting the promoter region, intron and exons?
Answer: We identified, in the Arapaima gigas growth hormone gene, four exons and three introns. their position in the genome assembly, at scaffold 1040, goes from nucleotide position 26175 to 25307 (introns: 25146-24367; 25670-25308; 25789-26038). This information was obtained from Gnomon pipeline, as mentioned in “in silico” prediction analysis, where the steps Blastp and SplingPro retrieve this exact position. Unfortunately, the chromosomal map for the genome demands an extended work. At this moment, we do not yet know on which chromosome the ag-GH gene is located.
4, 5 The figure1 showing the sequence of cDNA is not of high resolution, its resolution should be improved. The same problem of very low resolution is applied to figure 2
. Answer: We improved the resolution of Fig. 1 and Fig. 2 to 300 dpi and submitted them separately as figure files.
- 6. Table 4 is not clear what is included on it and why identity values repeated twice and of what the identity is? Perhaps the title of the table created a confusion
Answer: We agree and we changed the title of Table 3 to: “Percentage identity of GH peptides between fish orders” and of Table 4 to: “Percentage identity of GH peptides of different species relative to A. gigas or H. sapiens”.
- 7. Did you express this cDNA in E coli and check its molecular weight and western blot of ag-GH? Did the same approach also applied to extract this hormone from pituitary glands?
Answer: Yes, we are expressing ag-GH in E. coli and adding below the relative SDS-PAGE and Western blot analysis in comparison with hGH, using anti-hGH antiserum. The theoretical MW of ag-GH (21,150 Da) seems to be confirmed, but we will determine it exactly via MALDI-TOF-MS. We are also extracting ag-GH from A. gigas pituitary glands; however, to obtain and transport these glands from the northern part of Mato Grosso State (Piscicultura Raça) presents some practical difficulties. The data obtained will be the subject of a future work and would be practically impossible to include all of the data in the present paper.
|
SDS-PAGE and Western Blotting of the periplasmic fluid obtained with osmotic shock from BL21(DE3) pET3d-agGH cultivated at different temperatures with or without 0.4 mM IPTG. The molecular weight of hGH is 22129 Da and of agGH is 21150 Da. For the blotting anti-hGH antiserum was used.
- 8. Can you provide the agarose gel electrophoresis of both UTR and full length cDNA in the paper?
Answer: Yes, we can provide the agarose gel electrophoresis of the coding region together with the 3’ UTR; this sequence was deposited in the GenBank. We did not need the 5’ UTR for the present work, but we are working to obtain it.
- Molecular weight marker; 2. ~600 bp fragment corresponding to the coding sequence
- ~1000 bp fragment with the coding sequence + 3’UTR
- After amplification of whole coding sequences did you confirm the sequences with Sanger or NGS tool methods?
Answer: After amplification of the whole coding sequence, we confirmed it with Sanger sequencing, as now reported more clearly in Section 2.3.
- Line 91. Whats 100 MyR?
Answer: We followed the abbreviations used in paleontology, where x Myr refers to a period of x millions years, while x Ma indicates x millions years ago. We added this previously missing information in “Abbreviations”.
- 11. The paper need to be checked carefully for typos and grammatical error.
Answer: We checked the paper for typos and grammatical errors, thanks to a native English speaking Chemistry Professor.
Concerning novelty, we emphasized at the beginning of the Discussion that, for the first time, ag-GH cDNA has been sequenced and characterized and also an A. gigas hormone was sequenced via “in silico” methods, comparing this sequence with that obtained via the classical method. As far as we know, two homologous bioactive molecular structures, at an evolutionary distance of >100 millions of years, have never been compared by three-dimensional modelling and molecular dynamics. Considering Table 4, moreover, we paved the way for setting up a potentially useful new in vivo bioassay comparing the activities of ag-GH with hGH in a dwarf mouse model.

Reviewer 3 Report
To the author
Cloning of a cDNA homologous to several others without enough evidence for expression or functional analysis is preliminary. As the manuscript does not significantly increase our knowledge in the field, it does not seem to be appropriate for publishing. However, some comments for further improvement are proposed as follows:
1) The study does not define the gap in the field and the required fulfillment related to the presented data adequately. The last paragraph of the introduction and the first paragraph of the discussion should at least shed light on this fact.
2) The English Writing, Grammar, and Paragraphing have serious problems, which should be revised by a native language editor.
3) A schematic figure representing an image of the corresponding fish or embryo in the center and the signaling pathway of hormones is required.
4) The quality of sequences in figure 1 and figure 2 should be enhanced.
5) What is the usage of paragraph 4 on page 2 (Introduction): “The cloning and expression of the cDNA for salmon GH in E. coli was re-74 ported by Sekine et al. [21]. Similarly,………………..”.
Citing the work of others is valuable when you want to compare your results and draw a conclusion in favor or against your own results.
Author Response
The Authors thank the Reviewers for the interesting and very useful suggestions that contributed to a substantial improvement of the manuscript.
Reviewer # 3
Cloning of a cDNA homologous to several others without enough evidence for expression or functional analysis is preliminary. As the manuscript does not significantly increase our knowledge in the field, it does not seem to be appropriate for publishing. However, some comments for further improvement are proposed as follows:
1) The study does not define the gap in the field and the required fulfillment related to the presented data adequately. The last paragraph of the introduction and the first paragraph of the discussion should at least shed light on this fact.
Answer: We modified the last paragraph of the Introduction and the first paragraph of the Discussion in order to shed light on the points raised and on the fulfillment obtained. Since Reviewer # 2 expressed the same doubt concerning expression of ag-GH, we inform that in previous answer we sent SDS-PAGE and Western blotting analysis confirming bacterial ag-GH expression. These data can be added to the manuscript, if considered important. We are also purifying ag-GH in order to analyze it via MALDI-TOF-MS and determine its exact molecular weight.
2) The English Writing, Grammar, and Paragraphing have serious problems, which should be revised by a native language editor.
Answer: The present text has been revised by a native English speaking chemistry professor.
3) A schematic figure representing an image of the corresponding fish or embryo in the center and the signaling pathway of hormones is required.
Answer: A schematic figure representing an image of A. gigas and the signaling pathway of hormones has been added.
4) The quality of sequences in figure 1 and figure 2 should be enhanced.
Answer: The quality of Fig. 1 and Fig. 2 has been improved
5) What is the usage of paragraph 4 on page 2 (Introduction):“The cloning and expression of the cDNA for salmon GH in E.coli was reported by Sekine et al. [21].Similarly,………………..”.
Citing the work of others is valuable when you want to compare your results and draw a conclusion in favor or against your own results.
Answer: We agree with the Reviewer concerning paragraph 4 on page 2. We opted to leave only the citations, which are potentially of some utility to the reader interested in fish GH cDNA isolation and /or expression.
As already mentioned in the reply to Reviewer #2, at the end of the Introduction and at the beginning of the Discussion we emphasized the goals and the achievements of the work. For the first time ag-GH cDNA has been sequenced and characterized and also an A. gigas hormone has been sequenced via “in silico” methods, comparing this sequence with that obtained via the classical method. As far as we know, two homologous bioactive molecular structures, at an evolutionary distance of >100 millions of years, have never been compared by three-dimensional modelling and molecular dynamics. Considering Table 4, moreover, we paved the way for setting up a potentially useful new in vivo bioassay comparing the activities of ag-GH with hGH in a dwarf mouse model.

Reviewer 4 Report
Dear authors, the results support the principal objective and found the ag-GH sequence and made a 3D modeling to propose the protein that will be used for preliminary gene therapy studies. You have to make some modifications:
Include the web address of all servers and databases.
In table 2, the title of the last column is missing.
R226 in silico should be written in italics.
R227 A. gigas should be written in italics.
R244 Arapaima gigas should be written in italics.
R251 Arapaima gigas should be written in italics.
R266 Correct the title of Table 3.
In table 3, the text has white shading; remove it.
In table 4, the title of the last column is missing.
R251 Arapaima gigas should be written in italics.
R326 A. gigas should be written in italics.
In figure 3B, correct DeepRefiner.
Author Response
The Authors thank the Reviewers for the interesting and very useful suggestions that contributed to a substantial improvement of the manuscript.
Reviewer # 4
Dear authors, the results support the principal objective and found the ag-GH sequence and made a 3D modeling to propose the protein that will be used for preliminary gene therapy studies. You have to make some modifications:
1) Include the web address of all servers and databases.
Answer: We have included the web address of all servers.
2) In table 2, the title of the last column is missing.
Answer: In Table 2 we added the title of the last column
3) Correct the title of Table 3.
Answer: We corrected the titles of Table 3 and Table 4 that were indeed misleading
4) In table 3, the text has white shading; remove it.
Answer: In Table 3 the white shading has been removed
5) In table 4, the title of the last column is missing.
Answer: In Table 4 we added the title of the last column
6) When necessary all names have been written in italics

Round 2
Reviewer 2 Report
1-The authors has not appropriately addressed the issues for example, in table 4, i do not understand which identity value corresponding to homosapian and which one is to A. gigas, there is two column and the authors modified the title to the table.
2. The title of the paper is not modified.
3. The figures quality is still very very low.
4. There is no need to include a single abbreviations, you can just define it in bracket.
Author Response
Reviewer 2
1-The authors has not appropriately addressed the issues for example, in table 4, I do not understand which identity value corresponding to homosapian and which one is to A. gigas, there is two column and the authors modified the title to the table.
Answer: Table 4 is reporting the percentages identity between GH peptides (percent of identical amino acids) of different species at different evolution levels. Therefore, the percentages have been calculated like in Table 3, were the percentage identity is calculated just between fish orders. In the first column, 100% was considered the oldest species (A.gigas), while in the second column 100% was considered homo sapiens, as is usually done for practical reasons, in order to know how much a human peptide diverges from a less evolved species. We agree however that the table is somehow misleading; we eliminated therefore the second column, leaving the first one, which indicates divergence as it really occurred during evolution.
- The title of the paper is not modified.
Answer: The title was modified substituting “Molecular cloning” with “Isolation”, as we did in the title of Faria et al. (2013, Ref. 15).
- The figures quality is still very very low.
Answer: The quality of the figures is still particularly low because, sending them through the journal link via words, they loose quality. We sent therefore the original figures (300 DPI), without modifying them, via e-mail, directly to the Editor, Dr. Youssef Zhang.
- There is no need to include a single abbreviations, you can just define it in bracket.
Answer: The abbreviation was included into the text. See line 38: “>100 Myr evolutionary period (1Myr= 1 million of years)”

Reviewer 3 Report
The manuscript can be accepted considering revisions.
Author Response
Reviewer #3
Comments and Suggestions for Authors
The manuscript can be accepted considering revisions.